

# Novel association between TGFA, TGFB1, IRF1, PTGS2 and IKBKB single-nucleotide polymorphisms and occurrence, severity and treatment response of major depressive disorder

Katarzyna Bialek[1], Piotr Czarny[2], Cezary Watala[3], Paulina Wigner[1], Monika Talarowska[4], Piotr Galecki[5], Janusz Szemraj[2] and Tomasz Sliwinski[1]

[1] Laboratory of Medical Genetics, Faculty of Biology and Environmental Protection, University of Lodz, Lodz, Poland
[2] Department of Medical Biochemistry, Medical University of Lodz, Lodz, Poland
[3] Department of Haemostatic Disorders, Medical University of Lodz, Lodz, Poland
[4] Institute of Psychology, Department of Personality and Individual Differences, University of Lodz, Lodz, Poland
[5] Department of Adult Psychiatry, Medical University of Lodz, Lodz, Poland

Corresponding author
Tomasz Sliwinski,
tomasz.sliwinski@biol.uni.lodz.pl

## ABSTRACT

**Background:** Activation of the immune system might affect the severity of depressive episodes as well as response to the antidepressant treatment. The purpose of this study was to investigate whether the occurrence of variant alleles of analyzed SNPs are involved in prevalence and progression of depression. Moreover, selected genes and SNPs have not been investigated in context of the disease severity and treatment. Therefore, six polymorphisms were selected: g.41354391A>G-*TGFB1* (rs1800469), g.132484229C>A-*IRF* (rs2070729), g.186643058A>G-*PTGS2* (rs5275), g.186640617C>T-*PTGS2* (rs4648308), g.70677994G>A-*TGFA* (rs2166975) and g.42140549G>T–*IKBKB* (rs5029748).

**Methods:** A total of 360 (180 patients and 180 controls) DNA samples were genotyped using TaqMan probes.

**Results:** We observed that A/G of the rs2166975 *TGFA*, A/C of rs2070729 *IRF1* and G/T of rs5029748 *IKBKB* were associated with an increased risk of depression development while the T/T of rs5029748 *IKBKB*, T/T of rs4648308 *PTGS2* and G/G of rs2166975 *TGFA* reduced this risk. We also stratified the study group according to gender and found that genotype A/G and allele G of the rs2166975 *TGFA*, G/T of rs5029748 *IKBKB* as well as C allele of rs4648308 *PTGS2*, homozygote A/A and allele A of rs5275 *PTGS2* were associated with increased risk of depression development in men while homozygote G/G of rs5275 *PTGS2* decreased this risk. Moreover, C/T of rs4648308 *PTGS2* and A/G of rs5275 *PTGS2* was positively correlated with the risk of the disease occurrence in women. Furthermore, a gene–gene analysis revealed a link between studied polymorphisms and depression. In addition, A/A of rs1800469 *TGFB1* was associated with earlier age of onset of the disease while G/G of this SNP increased severity of the depressive episode. Interestingly, A/C of rs2070729 *IRF1* and T/T of rs5029748 *IKBKB* may modulate the

effectiveness of selective serotonin reuptake inhibitors therapy. In conclusion, studied SNPs may modulate the risk of occurrence, age of onset, severity of the disease and response to the antidepressant treatment.

## INTRODUCTION

Depression (Major depressive disorder, MDD) is one of the most frequently diagnosed mental diseases. According to World Health Organization, about 350 million people suffer from this disorder all over the world (*WHO, 2018*). Despite the importance of the problem, pathogenesis of depression is not fully understood. However, there is a growing body of evidence suggesting that immune system impairment and dysregulation is associated with the pathophysiology of MDD. In particular, the "cytokine hypothesis" is widely accepted as one of the mechanisms for the development of depression (*Capuron & Miller, 2011*). This theory postulates that MDD is a result of elevated expression of pro-inflammatory cytokines, which act as neuromodulators as well as main agents in mediation of the neuroendocrine, neurochemical and behavioral features of the disease (*Schiepers, Wichers & Maes, 2005*). Some evidence confirmed link between inflammation and depression. Primarily, MDD patients exhibit increased levels of cytokines and other pro-inflammatory markers (*Capuron & Miller, 2011*). Additionally, medical conditions connected with increased inflammatory response are associated with greater risk of MDD developing (*Capuron & Miller, 2011*).

One of the cytokine class strongly associated with depression are interferons (IFN), cluster of signaling proteins involved in immune response. More than twenty different IFN proteins have been identified so far and divided into classes. IFN proteins are able to activate immune cells, that is, natural killer cells (NK cells) and macrophages (*Pinto & Andrade, 2016*). For instance, IFN-α is implicated in modulation of mood, behavior and sleep-wake cycle, partially by its ability to activate the pro-inflammatory cytokine network including, interleukin 1 (IL-1), interleukin 6 (IL-6) and tumor necrosis factor alpha (TNF-α) (*Zahiu & Mihai, 2014*). IFN and IFN-inducible genes, involved in immunity and inflammation, are transcriptionally regulated by interferon regulatory factor 1 (IRF1) (*Tamura et al., 2008*). IRF1 was a first identified transcription factor in IFN system and as a member of interferon regulatory factor family, plays important role in controlling expression of aforementioned genes (*Kröger et al., 2002*). Besides this, IRF1 promotes inflammatory cytokine release and regulates expression of interleukin 12 (IL-12) and interleukin 15 (IL-15), which are involved in MDD (*Tamura et al., 2008*).

Besides cytokine theory, various inflammatory pathways are thought to be activated in course of depression, including activation of the NF-kB (nuclear factor-kB), what leads to increased levels of pro-inflammatory cytokines (*Bierhaus et al., 2003*; *Pace et al., 2006*). NF-kB is a ubiquitous transcriptional factor that regulates expression of genes involved in

pleiotropic functions, including pro-inflammatory cytokines and co-stimulatory molecules (*Takeda & Akira, 2007*; *Krakauer, 2008*; *Zhang, Lenardo & Baltimore, 2017*). Inactive NF-kB molecules retain in the cytoplasm by interaction with IkB proteins, allowing to immediate activation in response to adequate impulse (*Napetschnig & Wu, 2013*). Canonical signaling of NF-kB is activated by IkB kinase (IKK complex), consisting of three subunits, each encoded by separate gene, that is, IKK-a (Inhibitor of nuclear factor kappa-B kinase subunit alpha) encoded by *CHUK* gene, IKK-B (inhibitor of nuclear factor kappa-B kinase subunit beta) by *IKBKB* gene and IKK-g (inhibitor of nuclear factor kappa-B kinase subunit gamma) by *IKBKG*. The activation of IKK is induced by phosphorylation of serine residues in catalytic subunits of kinase complex (*Napetschnig & Wu, 2013*; *Karin & Ben-Neriah, 2000*; *Cardinez et al., 2018*). Therefore, defective expression of NF-kB as the pro-inflammatory transcription factor, caused by alterations in *IKBKB* gene, may play a role in the development of depression (*Napetschnig & Wu, 2013*).

Transforming growth factors (TGF) constitute of two classes of polypeptide growth factors, namely TGFA (transforming grow factor α) and TGFB (transforming grow factor β). Important functions of these cytokines are embryonic development and regulation of specific reactions of immune system by their ability to induce T regulatory cells (Treg) (*Kissin et al., 2002*; *Yamagiwa et al., 2001*). TGFA is a ligand for epidermal growth factor receptor, which stimulates cell migration and proliferation. These gene and protein have been associated with many types of cancers and other diseases (*Ten Dijke & Hill, 2004*). Another piece of evidence confirmed that TGFB, an anti-inflammatory cytokine, plays role in brain inflammation as well as in peripheral immune response. Namely, TGFB is mainly involved in regulating inflammatory response by induction of differentiation of CD4+ T cells (*Nam et al., 2008*; *Passos et al., 2010*). Another essential function of the protein is cell to cell signaling, and thus controlling of cell growth and differentiation (*Ten Dijke & Hill, 2004*). In addition, TGFB is able to exert neuroprotective effects in many neurodegenerative disorders (*Vivien & Ali, 2006*). Information about its role in depression are contradictory. On the one hand, in animal model of depression, the cytokine level is increased and causes imbalance between Treg and Th17 cells (*Hong et al., 2013*). On the other hand, some studies reported that levels of TGFB in depressed patients are lower than in healthy control group (*Musil et al., 2011*; *Sutcigil et al., 2007*). Moreover, TGFB alone is sufficient to stimulate production of pro-inflammatory cytokines for example, IL-1 and TNF-α (*Kunzmann et al., 2003*). The protein is also able to induce expression of prostaglandin-endoperoxide synthase 2 (PTGS2; cyclooxygenase-2—COX-2) encoded by *PTGS2* gene, which is involved in pathogenesis of MDD. PTGS2 besides contribution to processes related to inflammation, also participates in the production of free radicals, which is partly utilized by PTGS2 itself (*Aktan, 2004*; *Hansson, Olsson & Nauseef, 2006*). Moreover, COX-2 catalyzes conversion of arachidonic acid (AA) to prostaglandins (PGs), which further intensify inflammation and neurodegenerative processes in central nervous system (CNS) (*Minghetti, 2004*). In response to growth factors, cytokines and other inflammatory molecules, PTGS2 is immediately expressed and is responsible for the production of prostanoid in both acute and chronic inflammatory conditions

(*Breyer et al., 2001*; *Shi et al., 2010*). Additionally, in animal model of depression increased expression of PTGS2 was observed in brain regions (*Cassano et al., 2006*).

The evidence suggests that MDD may be associated with impairment of immune system, caused by defective activity of aforementioned genes. Moreover, genetic factors may play an essential role in development of depression, since genome-wide association studies (GWAS) found several regions significantly associated with MDD (*Shyn et al., 2011*; *Wray et al., 2018*). Therefore, the present study examines the prospective relationship between the occurrence, age of onset, severity or antidepressant treatment efficacy of MDD and appearance of single nucleotide polymorphism (SNP) located in inflammatory-related genes, that is, g.132484229C>A of *IRF1* (rs2070729, located on 5q31.1), g.186643058A>G of *PTGS2* (rs5275, located on 1q31.1), g.186640617C>T of *PTGS2* (rs4648308, located on 1q31.1), g.70677994G>A of *TGFA* (rs2166975, located on 2p13.3), g.41354391A>G of *TGFB1* (rs1800469, located on 19q13.2) and g.42140549G>T of *IKBKB* (rs5029748, located on 8p11.21). Selected SNPs are located within immune genes participating in inflammatory-related signaling pathways. Therefore, they could affect gene expression and protein function and thus contribute to immune disruptions leading to increased risk of MDD.

## MATERIALS AND METHODS

### Subjects

The study included a total of 360 participants randomly selected. A group of 180 patients with depression hospitalized at the Department of Adult Psychiatry of the Medical University of Lodz and 180 volunteers without health problems, selected randomly (Table 1). Participants who took part in the experiment were native, not-related Poles. Patients were included based on the criteria set out in ICD-10 (F32.0–7.32.2, F33.0–F33.8). Medical and psychiatric records were obtained in accordance with ICD-10 criteria, using the Standardized Composite International Diagnostic Interview (CIDI). The depression' severity was evaluated using the 21-item Hamilton Depression Rating Scale (HDRS-21). The exclusion criteria included: axis I and II disorders other than MDD, chronic somatic diseases, autoimmune disorders (psoriasis, rheumatoid arthritis, chronic obstructive pulmonary disease, cancer, chronic kidney disease, systemic lupus erythematosus, type 1 diabetes, hepatitis B and C virus and HIV infection), neuroinflammatory and neurodegenerative disorders (including multiple sclerosis, Alzheimer's disease, Parkinson's disease) and central nervous system damage. Furthermore, subjects with familial incidence of mental diseases, other than MDD did not participate in the experiment. Psychiatric examination was conducted by the same psychiatrist, before the subjects were included in the experiment and after 8 weeks of pharmacotherapy with selective serotonin reuptake inhibitor (SSRI). Control group included selected randomly, volunteers with negative history of mental disorders. Participation in the experiment was voluntary. Controls and patients who did not agree to participate in the study were excluded. The purpose of the study was clearly presented, participants were assured that their personal information would be kept confidential. All of the subjects agreed by giving

**Table 1 Characteristic of studied population.** M means male; F means Female Mdn—median; Q1—first quartile; Q3—third quartile HRDS1—points in Hamilton Depression Rating Scale measured before antidepressant treatment.

| Group | Control ($n = 180$) | Patients ($n = 180$) |
|---|---|---|
| Sex (M/F) | 93/87 | 91/89 |
| Age (Mdn ($Q_1$; $Q_3$)) | 57 (50; 65) | 51 (44; 56) |
| Age of onset (Mdn ($Q_1$; $Q_3$)) | – | 34 (28; 43) |
| HRDS1 (Mdn ($Q_1$; $Q_3$)) | – | 24 (19; 27) |
| Treatment efficacy | | |
| Responsive (reduction from baseline of ≥ 50% in the total score) | | 93% |
| Remission (total HRDS1 score ≤7) | | 66% |

**Table 2 Characteristic of studied polymorphisms.**

| Gene | rs number | Polymorphis | Localization | Minor allele freqency |
|---|---|---|---|---|
| *TGFA* | rs2166975 | g.70677994G>A | Exon 5 | $A = 0.256$ |
| *TGFB* | rs1800469 | g.41354391A>G | 5′ of *TGFB* gene | $A = 0.312$ |
| *IRF1* | rs2070729 | g.132484229C>A | Intron 9 | $A = 0.465$ |
| *IKBKB* | rs5029748 | g.42140549G>T | Intron 2 | $T = 0.259$ |
| *PTGS2* | rs5275 | g.186643058A>G | 3′ UTR of *PTGS2* gene | $G = 0.310$ |
| | rs4648308 | g.186640617C>T | 3′ of *PTGS2* gene | $T = 0.142$ |

their written consent to participate in the experiment according to the protocol approved by the Bioethics Committee of the Medical University of Lodz (No. RNN/70/14/KE).

## SNP selection

Selection of the studied polymorphisms was performed using the public domain of the database for single nucleotide polymorphisms of the National Center for Biotechnology (NCBI dbSNP, www.ncbi.nlm.nih.gov/snp/) (Bethesda, Montgomery County, MD, USA). The criteria used for the SNPs' selection were that the minor allele frequency is greater than 0.05 in the European population, and that they are located in the coding or regulatory region of the genes and may have functional meaning for transcription and protein function. Detailed information about selected polymorphisms are presented in Table 2.

## DNA isolation

Genomic DNA was isolated from venous blood in accordance with the manufacturer instructions. Blood samples were collected from control group and patients with MDD. Blood Mini Kit (A&A Biotechnology, Gdynia, Poland) was used to extract nucleic acid. The purity of and concentration of the DNA was measured spectrophotometrically by calculating the ratio between absorbance at 260 nm and 280 nm, using Picodrop™ (Picodrop Limited, Astranet Systems Ltd., Cambridge, UK). Samples were stored at −20 °C until use.

### Genotyping

The investigated SNPs were genotyped using a TaqMan SNP Genotyping Assay (Thermo Fisher Scientific, Waltham, MA, USA), and a 2X Master Mix Takyon for Probe Assay—No ROX (Eurogentec, Liège, Belgium). Reactions were conducted in accordance with the manufacturer's instruction. Real-time PCR were performed with a Bio-Rad CFX96 Real-Time PCR Detection System, and analyzed in CFX Manager Software (Bio-Rad Laboratories Inc., Hercules, CA, USA).

### Statistical analysis

The collected data were analyzed in Statistica 12 (Statsoft, Tulsa, OK, USA), SigmaPlot 14.0 (Systat Software Inc., San Jose, CA, USA), Resampling Stats Add-in for Excel v.4 (Arlington, TX, USA) and StudSize3.02 (CreoStat HB, Florunda, Sweden). The descriptive statistics are shown as medians with interquartile ranges. Normality of the studied group was verified with the Shapiro–Wilk test, homogeneity of variance was checked with Brown–Forsythe test. Accordingly, either the unpaired Student's $t$ test or Mann–Whitney $U$ test was used. To calculate the associations between studied polymorphisms and the occurrence of a disease an unconditional multiple logistic regression model was used. The results are shown as odds ratio (OR) with 95% confidence interval (95% CI). The OR values were adjusted for the potential confounders, including age and sex. We also stratified results into male and female group and evaluated correlation between case/control for each polymorphism. In addition, in order to strengthen that the revealed differences were not detected by a pure chance the significant outcomes were further validated with the use of two approaches: the bootstrap-boosted multiple logistic regression (resampling with replacement, 10,000 iterations) and the cross-validated logistic regression (corresponding to the $d$-jackknife technique), with the patient group acting as the modeled class. This was intended to overcome any possible bias related to relatively low sample sizes. The goodness of fit of logistic regression models showing a significant degree of discrimination between controls and patients was estimated with Hosmer–Lemeshow test.

Efficiency of the treatment was calculated using the formula as described before (*Czarny et al., 2019*):

$$TE = \frac{(HAM\text{-}D_0 - HAM\text{-}D_E) \times 100\%}{HAM\text{-}D_0}$$

TE-treatment efficiency; HAM-$D_0$—score before therapy; HAM-$D_E$—score after therapy.

## RESULTS

### Single nucleotide polymorphisms of genes encoding IRF1, IKBKB, TGFA, TGFB and PTGS2 as a risk of MDD

The distribution of genotypes and alleles in both depressed and control groups was in agreement with Hardy–Weinberg equilibrium. Results are presented in Table 3.
The results demonstrated that the A/G genotype of the g.70677994G>A (rs2166975)

**Table 3 Distribution of genotypes and alleles of rs1800469 (*TGFB1*), rs2070729 (*IRF1*), rs5275 (*PTGS2*), rs4648308 (*PTGS2*), rs2166975 (*TGFA*), rs5029748 (*IKBKB*) and the risk of depression occurrence.**

| Genotype/Allele | Control | | Depression | | Crude OR (95% CI) | p | Adjusted OR (95% CI)* | p |
|---|---|---|---|---|---|---|---|---|
| | Number | Frequency | Number | Frequency | | | | |
| g.41354391A>G of *TGFB1* (rs1800469) | | | | | | | | |
| A/A | 23 | 0.128 | 20 | 0.117 | 0.853 [0.451–1.616] | 0.626 | 0.739 [0.367–1.49] | 0.398 |
| A/G | 71 | 0.394 | 76 | 0.428 | 1.231 [0.623–2.432] | 0.550 | 1.197 [0.762–1.879] | 0.435 |
| G/G | 86 | 0.478 | 84 | 0.483 | 1.123 [0.575–2.196] | 0.734 | 0.949 [0.609–1.479] | 0.818 |
| $\chi^2 = 0.403$ ; $p = 0.818$ | | | | | | | | |
| A | 117 | 0.325 | 116 | 0.322 | 0.987 [0.723–1.349] | 0.937 | 0.961 [0.687–1.344] | 0.815 |
| G | 243 | 0.675 | 244 | 0.678 | 1.013 [0.741–1.384] | 0.937 | 1.041 [0.744–1.456] | 0.815 |
| g.70677994G>A of *TGFA* (rs2166975) | | | | | | | | |
| A/A | 27 | 0.142 | 15 | 0.081 | 0.530 [0.272–1.031] | 0.062 | 0.576 [0.280–1.184] | 0.133 |
| **A/G** | **59** | **0.311** | **83** | **0.446** | [b]**1.814 [1.197–2.749]** | **0.005** | [b]**2.115 [1.341–3.336]** | **0.001** |
| | | | | | 1.789 [1.173–2.728]$^{0.692}$ | 0.007 | 2.091 [1.323–3.304]$^{0.893}$ | 0.002 |
| **G/G** | **104** | **0.547** | **88** | **0.473** | 0.743 [0.495–1.114] | 0.150 | [b]**0.609 [0.392–0.946]** | **0.027** |
| | | | | | | | 0.615 [0.395–0.957]$^{0.691}$ | 0.031 |
| $\chi^2 = 8.627$ ; $p = 0.013$ | | | | | | | | |
| A | 113 | 0.297 | 113 | 0.304 | 1.031 [0.755–1. 408] | 0.848 | 1.173 [0.839–1.640] | 0.351 |
| G | 267 | 0.703 | 259 | 0.696 | 0.970 [0.710–1.325] | 0.848 | 0.853 [0.610–1.192] | 0.351 |
| g.132484229C>A of *IRF1* (rs2070729) | | | | | | | | |
| A/A | 37 | 0.209 | 36 | 0.193 | 0.902 [0.540–1.507] | 0.694 | 0.883 [0.507–1.539] | 0.661 |
| **A/C** | **76** | **0.429** | **99** | **0.529** | [b]**1.409 [1.002–2.216]** | **0.048** | [b]**1.504 [0.963–2.348]** | **0.077** |
| | | | | | 1.495 [0.989–2.261]$^{0.457}$ | 0.057 | 1.496 [0.957–2.337] | 0.073 |
| C/C | 64 | 0.362 | 52 | 0.278 | 0.680 [0.437–1.059] | 0.088 | 0.692 [0.429–1.115] | 0.130 |
| $\chi^2 = 4.006$; $p = 0.135$ | | | | | | | | |
| A | 150 | 0.424 | 171 | 0.457 | 1.146 [0.855–1.536] | 0.363 | 1.225 [0.893–1.681] | 0.208 |
| C | 204 | 0.576 | 203 | 0.543 | 0.873 [0.651–1.170] | 0.363 | 0.816 [0.595–1.120] | 0.208 |
| g.42140549G>T of *IKBKB* (rs5029748) | | | | | | | | |
| G/G | 108 | 0.587 | 100 | 0.559 | 0.891 [0.588–1.350] | 0.586 | 0.928 [0.594–1.450] | 0.743 |
| **G/T** | **40** | **0.217** | **59** | **0.330** | [b]**1.787 [1.125–2.839]** | **0.014** | [b]**1.813 [1.072–3.066]** | **0.026** |
| | | | | | 1.770 [1.108–2.829]$^{0.551}$ | 0.017 | 1.776 [1.080–2.921]$^{0.556}$ | 0.024 |
| **T/T** | **36** | **0.196** | **20** | **0.112** | [b]**0.507 [0.272–0.945]** | **0.032** | [b]**0.450 [0.229–0.885]** | **0.021** |
| | | | | | 0.517 [0.286–0.934]$^{0.647}$ | 0.029 | 0.461 [0.243–0.877]$^{0.759}$ | 0.018 |
| $\chi^2 = 1.509$; $p = 0.470$ | | | | | | | | |
| G | 256 | 0.696 | 259 | 0. 723 | 1.145 [0.830–1.578] | 0.409 | 1.210 [0.857–1.707] | 0.279 |
| T | 112 | 0.304 | 99 | 0.277 | 0.874 [0.634–1.204] | 0.409 | 0.827 [0.586–1.167] | 0.279 |
| g.186643058A>G of *PTGS2* (rs5275) | | | | | | | | |
| A/A | 79 | 0.422 | 81 | 0.433 | 1.045 [0.693–1.574] | 0.834 | 1.079 [0.696–1.674] | 0.734 |
| A/G | 75 | 0.401 | 83 | 0.444 | 1.192 [0.790–1.797] | 0.402 | 1.262 [0.812–1.961] | 0.302 |
| G/G | 33 | 0.176 | 23 | 0.123 | 0.654 [0.368–1.164] | 0.149 | 0.550 [0.295–1.024] | 0.059 |
| $\chi^2 = 1.848$; $p = 0.397$ | | | | | | | | |
| A | 233 | 0.623 | 245 | 0.655 | 1.149 [0.853–1.549] | 0.361 | 1.225 [0.890–1.688] | 0.214 |
| G | 141 | 0.377 | 129 | 0.345 | 0.870 [0.675–1.173] | 0.361 | 0.816 [0.593–1.124] | 0.214 |

(Continued)

| Genotype/Allele | Control | | Depression | | Crude OR (95% CI) | p | Adjusted OR (95% CI)* | p |
|---|---|---|---|---|---|---|---|---|
| | Number | Frequency | Number | Frequency | | | | |
| g.186640617C>T of *PTGS2* (rs4648308) | | | | | | | | |
| C/C | 130 | 0.703 | 124 | 0.697 | 0.972 [0.620–1.522] | 0.900 | 0.927 [0.575–1.496] | 0.756 |
| C/T | 40 | 0.216 | 52 | 0.292 | 1.496 [0.929–2.409] | 0.097 | [b]1.673 [0.994–2.815] | **0.052** |
| | | | | | | | 1.650 [0.991–2.745][0.438] | **0.054** |
| **T/T** | **14** | **0.076** | **2** | **0.011** | [b]**0.129 [0.027–0.631]** | **0.011** | [b]**0.103 [0.029–0.511]** | **0.003** |
| | | | | | 0.139 [0.031–0.620][0.932] | 0.010 | 0.110 [0.023–0.522][0.946] | 0.005 |
| $\chi^2$ =10.61; p = 0.005 | | | | | | | | |
| C | 300 | 0.815 | 300 | 0.843 | 1.2148 [0.824–1.790] | 0.327 | 1.208 [0.799–1.828] | 0.370 |
| T | 68 | 0.184 | 56 | 0.157 | 0.824 [0.559–1.214] | 0.327 | 0.828 [0.547–1.252] | 0.370 |

Notes:
* 'Adjusted OR' means OR adjusted for sex and age; for significant comparisons the superscript b means the bootstrap-boosted OR (resampling with replacement, 10,000 iterations); all OR values without bootstrap analysis were calculated using cross-validation algorithm.
Statistical power $(1 - \beta)$ (calculated at $\alpha = 0.05$) for significant comparisons given in superscripts.
$p < 0.05$ along with corresponding ORs are in bold.

polymorphism of the *TGFA* gene is associated with an increased risk of depression development, while G/G genotype decreased this risk. Furthermore, in case of *IRF1*, carriers of A/C genotype of the g.132484229C>A (rs2070729) have a greater chance of developing the disease. Moreover, the T/T homozygote of g.186640617C>T (rs4648308) of *PTGS2* gene is negatively correlated with risk of MDD development. Similarly, In the case of g.42140549G>T (rs5029748) polymorphism of *IKBKB*, we found that T/T homozygote decreased risk of MDD occurrence, while the heterozygote of the same gene variant decreased this risk.

## Single-nucleotide polymorphisms of genes encoding IRF1, IKBKB, TGFA, TGFB and PTGS2 and MDD occurrence in male and female population

Since women show two-times higher risk of MDD occurrence compared to men, we decided to investigated the association between prevalence of the disease in stratified male/female population and all studied SNPs. Results are presented in Table 4. The results demonstrated that in the case of g.70677994G>A (rs2166975) polymorphism of the *TGFA*, the A/G genotype increased the risk of MDD in men, but not in women. Moreover, allele A of this SNP was associated with decreased chance of the disease, while allele G was strongly correlated with higher risk of MDD. Furthermore, in male population allele G and G/G homozygote of the g.186643058A>G (rs5275) of *PTGS2* decreased risk of depression while, allele A and A/A homozygote of the same polymorphism was associated with increased risk of the occurrence of the disease. Additionally, it was found that A/G genotype of this SNP was correlated with higher risk of MDD in the female group. Another SNP of *PTGS2* gene, g.186640617C>T (rs4648308) was associated with MDD risk in both studied groups. Precisely, C/T genotype was positively correlated with the risk of the occurrence of MDD in women. Similarly, allele C of the mentioned polymorphism
**Table 4 Distribution of genotypes and alleles of rs2070729 (*IRF1*), rs5275 (*PTGS2*), rs4648308 (*PTGS2*), rs2166975 (*TGFA*), rs5029748 (*IKBKB*) and the risk of depression occurrence in male and female population.**

| Genotype/Allele | Control | | Depression | | Crude OR (95% CI) | p | Adjusted OR (95% CI)* | p |
|---|---|---|---|---|---|---|---|---|
| | Number | Frequency | Number | Frequency | | | | |
| **Male** | | | | | | | | |
| g.70677994G>A of *TGFA* (rs2166975) | | | | | | | | |
| A/A | 13 | 0.126 | 8 | 0.085 | 0.644 [0.254–1.641] | 0.353 | 0.808 [0.302–2.164] | 0.671 |
| **A/G** | **34** | **0.330** | **45** | **0.479** | [b]1.843 [1.037–3.272] | 0.037 | [b]2.318 [1.222–4.400] | 0.010 |
| | | | | | 1.864 [1.047–3.317][0.468] | 0.034 | 2.280 [1.218–4.268][0.733] | 0.009 |
| G/G | 56 | 0.544 | 41 | 0.436 | 0.649 [0.370–1.145] | 0.132 | [b]0.476 [0.250–0.905] | 0.024 |
| | | | | | | | 0.480 [0.257–0.898][0.740] | 0.022 |
| $\chi^2 = 4.640$; $p = 0.098$ | | | | | | | | |
| A | 60 | 0.291 | 61 | 0.709 | [b]0.109 [0.062–0.189] | <0.001 | [b]0.106 [0.060–0.186] | <0.001 |
| | | | | | 0.113 [0.065–0.195][0.992] | <0.001 | 0.109 [0.063–0.190][0.999] | <0.001 |
| G | 146 | 0.324 | 127 | 0.676 | [b]9.005 [5.242–15.468] | <0.001 | [b]9.281 [5.308–16.225] | <0.001 |
| | | | | | 8.861 [5.125–15.319][0.992] | <0.001 | 9.135 [5.260–15.867][0.999] | <0.001 |
| g.42140549G>T of *IKBKB* (rs5029748) | | | | | | | | |
| G/G | 61 | 0.610 | 53 | 0.589 | 0.916 [0.510–1.644] | 0.769 | 0.955 [0.514–1.776] | 0.885 |
| **G/T** | **19** | **0.190** | **30** | **0.333** | [b]2.153 [1.082–4.288] | 0.029 | [b]2.073 [1.016–4.300] | 0.049 |
| | | | | | 2.132 [1.097–4.143][0.466] | 0.026 | 2.063 [1.024–4.154][0.423] | 0.049 |
| **T/T** | **20** | **0.200** | **7** | **0.078** | [b]0.316 [0.118–0.849] | 0.022 | [b]0.295 [0.100–0.869] | 0.027 |
| | | | | | 0.337 [0.135–0.841][0.758] | 0.020 | 0.310 [0.116–0.830][0.799] | 0.021 |
| $\chi^2 = 8.788$; $p = 0.012$ | | | | | | | | |
| G | 141 | 0.705 | 136 | 0.756 | 1.211 [0.817–1.796] | 0.341 | 1.305 [0.826–2.061] | 0.253 |
| T | 59 | 0.295 | 44 | 0.244 | 0.826 [0.557–1.225] | 0.341 | 0.766 [0.485–1.210] | 0.253 |
| g.186643058A>G of *PTGS2* (rs5275) | | | | | | | | |
| **A/A** | **40** | **0.392** | **48** | **0.505** | 1.583 [0.896–2.796] | 0.111 | [b]2.073 [0.999–4.300] | 0.050 |
| | | | | | | | 1.803 [0.982–3.309][0.464] | 0.057 |
| A/G | 41 | 0.402 | 37 | 0.389 | 0.949 [0.534–1.687] | 0.858 | 0.852 [0.462–1.575] | 0.611 |
| G/G | 21 | 0.206 | 10 | 0.105 | 0.454 [0.201–1.028] | 0.057 | [b]0.427 [0.171–1.019] | 0.052 |
| | | | | | | | 0.438 [0.186–1.032][0.599] | 0.059 |
| $\chi^2 = 4.593$; $p = 0.101$ | | | | | | | | |
| **A** | **121** | **0.593** | **133** | **0.700** | [b]1.588 [1.031–2.445] | 0.036 | [b]1.659 [1.064–2.586] | 0.025 |
| | | | | | 1.601 [1.054–2.430][0.672] | 0.027 | 1.664 [1.087–2.548][0.745] | 0.019 |
| **G** | **83** | **0.407** | **57** | **0.300** | [b]0.621 [0.399–0.968] | 0.035 | [b]0.603 [0.393–0.926] | 0.021 |
| | | | | | 0.625 [0.412–0.949][0.608] | 0.027 | 0.601 [0.393–0.920][0.666] | 0.019 |
| g.186640617C>T of *PTGS2* (rs4648308) | | | | | | | | |
| C/C | 65 | 0.663 | 67 | 0.736 | 1.417 [0.754–2.664] | 0.276 | 1.335 [0.687–2.595] | 0.394 |
| C/T | 25 | 0.255 | 24 | 0.264 | 1.046 [0.543–2.014] | 0.892 | 1.128 [0.564–2.255] | 0.734 |
| T/T | 8 | 0.082 | 0 | 0 | – | – | – | – |
| $\chi^2 = 7.802$; $p = 0.020$ | | | | | | | | |
| **C** | **155** | **0.791** | **158** | **0.868** | [b]1.772 [0.996–3.162] | 0.052 | [b]1.744 [0.983–3.094] | 0.049 |
| | | | | | 1.741 [1.004–3.019][0.848] | 0.048 | 1.751 [1.007–3.040][0.854] | 0.047 |
| **T** | **41** | **0.209** | **24** | **0.132** | [b]0.567 [0.322–0.996] | 0.049 | [b]0.566 [0.315–0.999] | 0.049 |
| | | | | | 0.574 [0.331–0.996][0.553] | 0.048 | 0.571 [0.329–0.993][0.558] | 0.047 |

(Continued)

| Genotype/Allele | Control | | Depression | | Crude OR (95% CI) | $p$ | Adjusted OR (95% CI)* | $p$ |
|---|---|---|---|---|---|---|---|---|
| | Number | Frequency | Number | Frequency | | | | |
| **Female** | | | | | | | | |
| g.132484229C>A of *IRF1* (rs2070729) | | | | | | | | |
| A/A | 20 | 0.244 | 20 | 0.215 | 0.849 [0.417–1.730] | 0.650 | 0.655 [0.297–1.437] | 0.291 |
| **A/C** | **32** | **0.390** | **48** | **0.516** | **1.667 [0.910–3.056]** | **0.096** | [b]**2.016 [1.025–3.966]** | **0.042** |
| | | | | | | | **1.936 [1.003–3.738]**[0.508] | **0.049** |
| C/C | 30 | 0.366 | 25 | 0.269 | 0.637 [0.334–1.216] | 0.169 | 0.657 [0.328–1.318] | 0.237 |
| $\chi^2 = 2.975$; $p = 0.226$ | | | | | | | | |
| A | 72 | 0.439 | 88 | 0.473 | 1.136 [0.756–1.706] | 0.539 | 1.159 [0.730–1.840] | 0.532 |
| C | 92 | 0.561 | 98 | 0.527 | 0.880 [0.586–1.322] | 0.539 | 0.862 [0.544–1.370] | 0.532 |
| g.186643058A>G of *PTGS2* (rs5275) | | | | | | | | |
| A/A | 39 | 0.459 | 33 | 0.359 | 0.661 [0.359–1.211] | 0.176 | 0.595 [0.309–1.143] | 0.119 |
| **A/G** | **34** | **0.400** | **46** | **0.500** | **1.500 [0.823–2.734]** | **0.183** | **1.962 [1.024–3.758]** | **0.042** |
| | | | | | | | **1.952 [1.017–3.746]**[0.524] | **0.044** |
| G/G | 12 | 0.141 | 13 | 0.141 | 1.001 [0.427–2.344] | 0.998 | 0.718 [0.284–1.812] | 0.483 |
| $\chi^2 = 2.006$; $p = 0.356$ | | | | | | | | |
| A | 112 | 0.659 | 112 | 0.609 | 0.810 [0.527–1.244] | 0.336 | 0.833 [0.524–1.325] | 0.441 |
| G | 58 | 0.341 | 72 | 0.391 | 1.235 [0.804–1.898] | 0.336 | 0.816 [0.755–1.908] | 0.441 |
| g.186640617C>T of *PTGS2* (rs4648308) | | | | | | | | |
| C/C | 65 | 0.756 | 57 | 0.655 | 0.614 [0.315–1.195] | 0.148 | 0.595 [0.294–1.205] | 0.149 |
| **C/T** | **15** | **0.174** | **28** | **0.322** | [b]**2.270 [1.100–4.684]** | **0.027** | [b]**2.574 [1.224–5.415]** | **0.013** |
| | | | | | **2.246 [1.098–4.596]**[0.806] | **0.027** | **2.533 [1.178–5.449]**[0.587] | **0.017** |
| T/T | 6 | 0.070 | 2 | 0.023 | 0.314 [0.061–1.620] | 0.163 | 0.211 [0.037–1.211] | 0.081 |
| $\chi^2 = 6.449$; $p = 0.039$ | | | | | | | | |
| C | 145 | 0.843 | 142 | 0.816 | 0.843 [0.495–1.435] | 0.530 | 0.853 [0.469–1.553] | 0.603 |
| T | 27 | 0.157 | 32 | 0.184 | 1.186 [0.697–2.018] | 0.530 | 1.172 [0.644–2.135] | 0.603 |

Notes:

* 'Adjusted OR' means OR adjusted for sex and age; for significant comparisons the superscript b means the bootstrap-boosted OR (resampling with replacement, 10,000 iterations); all OR values without bootstrap analysis were calculated using cross-validation algorithm.
Statistical power $(1 - \beta)$ (calculated at $\alpha = 0.05$) for significant comparisons given in superscripts.
$p < 0.05$ along with corresponding ORs are in bold.

increased prevalence of the disease among men, while allele T decreased this risk. We also found that genotypes of g.42140549G>T (rs5029748) polymorphism of *IKBKB* gene were related with appearance of MDD in male population. Particularly, G/T genotype was connected with increased risk of depression, while T/T genotype of the same SNP decreased this risk.

## Gene-gene interactions of IRF1, IKBKB, TGFA, TGFB and PTGS2 and the risk of MDD

In this research, we also studied whether the combined genotypes of investigated polymorphism are associated with appearance of MDD. Results are presented in Table 5. In reference to effect of combined genotypes, it was found that G/G-T/T genotypes of

**Table 5 Gene-gene interactions of rs1800469 (*TGFB1*), rs2070729 (*IRF1*), rs5275 (*PTGS2*), rs4648308 (*PTGS2*), rs2166975 (*TGFA*), rs5029748 (*IKBKB*) and the risk of depression occurrence.**

| Combined genotype | Control (*n* = 180) | | Depression (*n* = 180) | | Crude OR (95% CI) | *p* | Adjusted OR (95% CI)* | *p* |
|---|---|---|---|---|---|---|---|---|
| | Number | Frequency | Number | Frequency | | | | |
| g.41354391A>G of *TGFB1* (rs1800469)–g.70677994G>A of *TGFA* (rs2166975) | | | | | | | | |
| A/G-A/G | 24 | 0.126 | 35 | 0.186 | 1.592 [0.904–2.803] | 0.106 | [b]1.906 [1.032–3.518] | 0.039 |
| | | | | | | | 1.898 [1.036–3.477]$^{0.490}$ | 0.038 |
| g.70677994G>A of *TGFA* (rs2166975)–g.132484229C>A of *IRF1* (rs2070729) | | | | | | | | |
| A/G-A/C | 27 | 0.141 | 45 | 0.241 | [b]1.951 [1.152–3. 305] | 0.013 | [b]2.117 [1.224–3.660] | 0.007 |
| | | | | | 1.925 [1.136–3.262]$^{0.554}$ | 0.015 | 2.092 [1.193–3.660]$^{0.667}$ | 0.010 |
| g.70677994G>A of *TGFA* (rs2166975)–g.186643058A>G of *PTGS2* (rs5275) | | | | | | | | |
| G/G-G/G | 22 | 0.115 | 8 | 0.043 | [b]0.320 [0.127–0.807] | 0.016 | [b]0.223 [0.087–0.574] | 0.002 |
| | | | | | 0.341 [0.148–0.788]$^{0.828}$ | 0.012 | 0.233 [0.094–0.579]$^{0.940}$ | 0.002 |
| A/A-G/G | 7 | 0.037 | 1 | 0.005 | 0.139 [0.017–1.141] | 0.066 | [b]0.167 [0.027–1.031] | 0.054 |
| | | | | | 0.129 [0.014–1.159]$^{0.805}$ | 0.068 |
| A/G-G/G | 4 | 0.021 | 14 | 0.074 | [b]3.581 [1.233–13.12] | 0.026 | [b]4.264 [1.416–12.839] | 0.010 |
| | | | | | 3.761 [1.215–11.647]$^{0.291}$ | 0.022 | 4.137 [1.263–13.545]$^{0.291}$ | 0.019 |
| g.70677994G>A of *TGFA* (rs2166975)–g.186640617C>T of *PTGS2* (rs4648308) | | | | | | | | |
| G/G-T/T | 12 | 0.063 | 1 | 0.005 | [b]0.087 [0.013–0.638] | 0.018 | [b]0.057 [0.011–0.312] | 0.001 |
| | | | | | 0.080 [0.010–0.620]$^{0.942}$ | 0.016 | 0.051 [0.006–0.420]$^{0.948}$ | 0.006 |
| A/G-C/T | 10 | 0.052 | 25 | 0.133 | [b]3.005 [1.242–7.269] | 0.015 | [b]3.240 [1.442–7.280] | 0.004 |
| | | | | | 2.776 [1.294–5.956]$^{0.584}$ | 0.009 | 3.115[1.397–6.944] $^{0.663}$ | 0.005 |
| g.70677994G>A of *TGFA* (rs2166975)–g.42140549G>T of *IKBKB* (rs5029748) | | | | | | | | |
| G/G-T/T | 23 | 0.120 | 9 | 0.048 | [b]0.362 [0.156–0.840] | 0.018 | [b]0.286 [0.106–0.772] | 0.013 |
| | | | | | 0.367 [0.165–0.816]$^{0.801}$ | 0.014 | 0.306 [0.131–0.719]$^{0.882}$ | 0.007 |
| A/G-G/T | 11 | 0.058 | 24 | 0.128 | [b]2.393 [1.136–5.042] | 0.022 | [b]2.645 [1.184–5.910] | 0.018 |
| | | | | | 2.395 [1.138–5.041]$^{0.472}$ | 0.021 | 2.621 [1.208–5.688]$^{0.571}$ | 0.015 |
| g.132484229C>A of *IRF1* (rs2070729)–g.186643058A>G of *PTGS2* (rs5275) | | | | | | | | |
| A/C-A/G | 29 | 0.152 | 49 | 0.261 | [b]2.077 [1.206–3.576] | 0.008 | [b]1.863 [1.022–3.394] | 0.042 |
| | | | | | 1.969 [1.180–3.286]$^{0.614}$ | 0.009 | 1.844 [1.069–3.180]$^{0.515}$ | 0.028 |
| g.132484229C>A of *IRF1* (rs2070729)–g.42140549G>T of *IKBKB* (rs5029748) | | | | | | | | |
| A/C-G/T | 16 | 0.084 | 29 | 0.154 | [b]2.032 [1.036–3.989] | 0.039 | [b]1.918 [0.935–3.931] | 0.075 |
| | | | | | 1.995 [1.044–3.810]$^{0.402}$ | 0.036 | 1.901 [0.958–3.774]$^{0.362}$ | 0.066 |
| g.42140549G>T of *IKBKB* (rs5029748)–g.186643058A>G of *PTGS2* (rs5275) | | | | | | | | |
| T/T-G/G | 14 | 0.073 | 2 | 0.011 | [b]0.131 [0.037–0.598] | 0.008 | [b]0.126 [0.027–0.589] | 0.008 |
| | | | | | 0.136 [0.030–0.607]$^{0.936}$ | 0.009 | 0.132 [0.02–0.610]$^{0.939}$ | 0.009 |
| G/T-A/G | 16 | 0.084 | 31 | 0.165 | [b]2.235 [1.114–4.487] | 0.024 | [b]1.933 [0.883–4.233] | 0.008 |
| | | | | | 2.160 [1.138–4.098]$^{0.512}$ | 0.018 | 1.894 [0.968–3.704]$^{0.357}$ | 0.009 |
| g.42140549G>T of *IKBKB* (rs5029748)–g.186640617C>T of *PTGS2* (rs4648308) | | | | | | | | |
| G/T-C/T | 4 | 0.021 | 19 | 0.101 | [b]5.013 [1.531–18.121] | 0.005 | [b]4.164 [1.232–15.343] | 0.035 |
| | | | | | 5.256 [1.753–15.760]$^{0.291}$ | 0.003 | 4.320 [1.390–13.428]$^{0.286}$ | 0.011 |

**Notes:**
* 'Adjusted OR' means OR adjusted for sex and age; for significant comparisons the superscript b means the bootstrap-boosted OR (resampling with replacement, 10,000 iterations); all OR values without bootstrap analysis were calculated using cross-validation algorithm.
Statistical power (1 − β) (calculated at α = 0.05) for significant comparisons given in superscripts.
*p* < 0.05 along with corresponding ORs are in bold.

g.70677994G>A (rs2166975)—*TGFA* and g.186640617C>T (rs4648308)—*PTGS2* was associated with decreased risk of depression occurrence, while A/G-C/T genotypes increased this risk. The A/G-A/C genotypes of g.70677994G>A (rs2166975)—*TGFA* and g.132484229C>A (rs2070729)—*IRF1* as well as A/G-A/G genotypes of g.70677994G>A (rs2166975) *TGFA* and g.41354391A>G (rs1800469)—*TGFB* also increased the risk of the disease. Furthermore, higher risk of MDD occurrence was associated with the G/T-A/G genotypes of g.42140549G>T (rs5029748)—*IKBKB* and g.186643058A>G (rs5275)—*PTGS2*, however the T/T-G/G genotypes reduced this risk. In the case of linked genotypes of g.70677994G>A (rs2166975)—*TGFA* and g.186643058A>G (rs5275)—*PTGS2*, we found that link between A/G-G/G of this genes was associated with higher risk of appearance of the MDD, while G/G-G/G as well as A/A-G/G genotypes decreased this chance. Similarly, A/G-G/T combined genotypes of g.70677994G>A (rs2166975)—*TGFA* and g.42140549G>T (rs5029748)—*IKBKB* increased risk of MDD but G/G-T/T genotypes of the same SNP were associated with lower risk of disease incidence. Moreover, carriers of A/C-A/G combined genotypes of g.132484229C>A (rs2070729)—I*RF1* and g.186643058A>G (rs5275)—*PTGS*, A/C-G/T of g.132484229C>A (rs2070729)—I*RF1* and g.42140549G>T (rs5029748)—*IKBKB* as well as G/T-C/T genotypes of g.42140549G>T (rs5029748)—*IKBKB* and g.186640617C>T (rs4648308)—*PTGS2* had a greater risk of MDD appearance.

## Single-nucleotide polymorphisms of genes encoding IRF1, IKBKB, TGFA, TGFB, PTGS2 and the age of the first episode of MDD and the severity classification on the hamilton depression rating scale

To estimate whether the investigated polymorphisms may had an impact on the age of the first episode of MDD, patients were stratified in accordance to genotype and their age of onset was compared (Fig. 1). A significant difference was found between A/A and A/G genotypes as well as A/A and G/G genotypes of g.41354391A>G (rs1800469)—*TGFB1*. Carriers of A/A genotype had their first episode significantly earlier compared to other genotypes.

In the case of the impact of genotypes of the investigated SNPs on the episode severity measured using the Hamilton Depression Rating Scale (HDRS) (Fig. 2), significant differences was found between carriers of A/A and G/G genotypes of g.41354391A>G (rs1800469)—*TGFB1*.

## Single-nucleotide polymorphisms of genes encoding IRF1, IKBKB, TGFA, TGFB, PTGS2 and effectiveness of depression treatment

We also evaluated impact of the studied polymorphisms on the effectiveness of antidepressant treatment with selective serotonin reuptake inhibitor (SSRI) (Fig. 3). Regarding the effect of investigated SNPs on treatment efficiency, differences was found between A/A and A/C genotypes of g.132484229C>A (rs2070729)—*IRF1* as well as G/G and T/T genotypes of g.42140549G>T (rs5029748)—*IKBKB*.

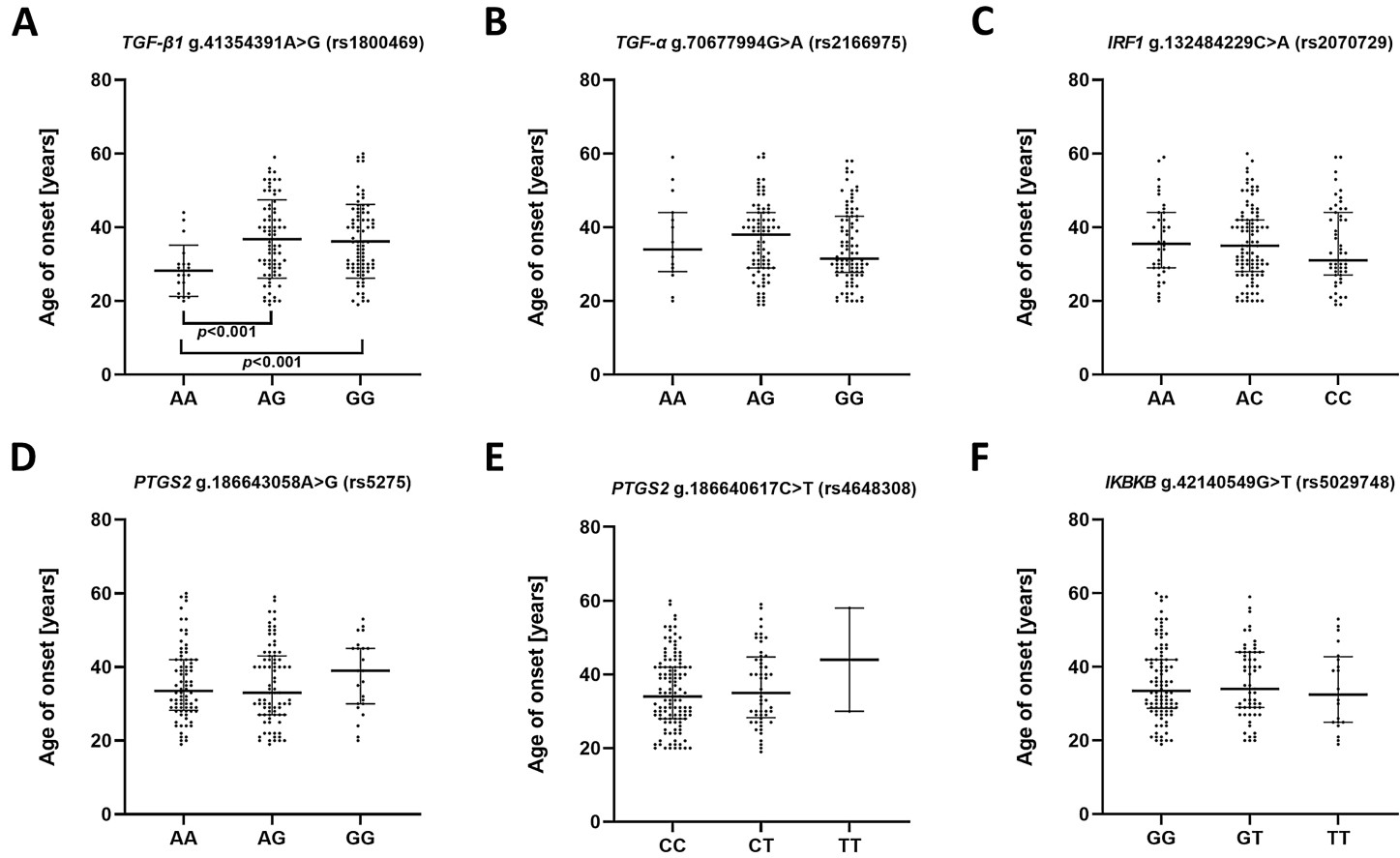

**Figure 1 Impact of single-nucleotide polymorphisms localized in inflammatory genes on the age of the first episode of MDD.** (A) *TGFB1* g.41354391A>G (rs1800469) (B) *TGFA* g.70677994G>A (rs2166975) (C) *IRF1* g.132484229C>A (rs2070729) (D) *PTGS2* g.186643058A>G (rs5275) (E) *PTGS2* g.186640617C>T (rs4648308) (F) *IKBKB* g.42140549G>T (rs5029748). Results are presented as scatter dot plots. The horizontal lines denote the median, while the whiskers show the inter-quartile range.                     

## DISCUSSION

There is strong amount of evidence that inflammation is undeniably associated with major depressive disorder. Moreover, it was confirmed that some inflammatory genes and presence of their genetics variants play important role in MDD development. Additionally, several loci/chromosomal regions connected with MDD were mapped by genome-wide linkage analysis, that is, 1q32.1, 2p25.1, 3p21.1, 3p26.1, 3q26.1, 6p22.3, 8q22.2, 8q22.3, 8q12.1, 8q23.3, 11p14.2-p14.3, 13q31.1–q31.3, 15q25.2 and 19q12 (*McGuffin et al., 2005*; *Shyn et al., 2011*; *Sullivan et al., 2013*). Selected candidate genes in current study are located in proximity to the above mentioned regions of chromosomes. In this research, we genotyped six polymorphic variants of *TGFA, TGFB1, IRF1* and *PTGS2* genes; and to our knowledge, none of this SNPs have been studied in the context of severity and treatment response in depression before. However, these SNPs were included in GWAS but only one of them, that is, rs2070729, had $p$ value below 0.05.

The first of investigated polymorphisms in this study was g.70677994G>A (rs2166975)— *TGFA*. The SNP is localized on 2p13.3 and it is responsible for synonymous change

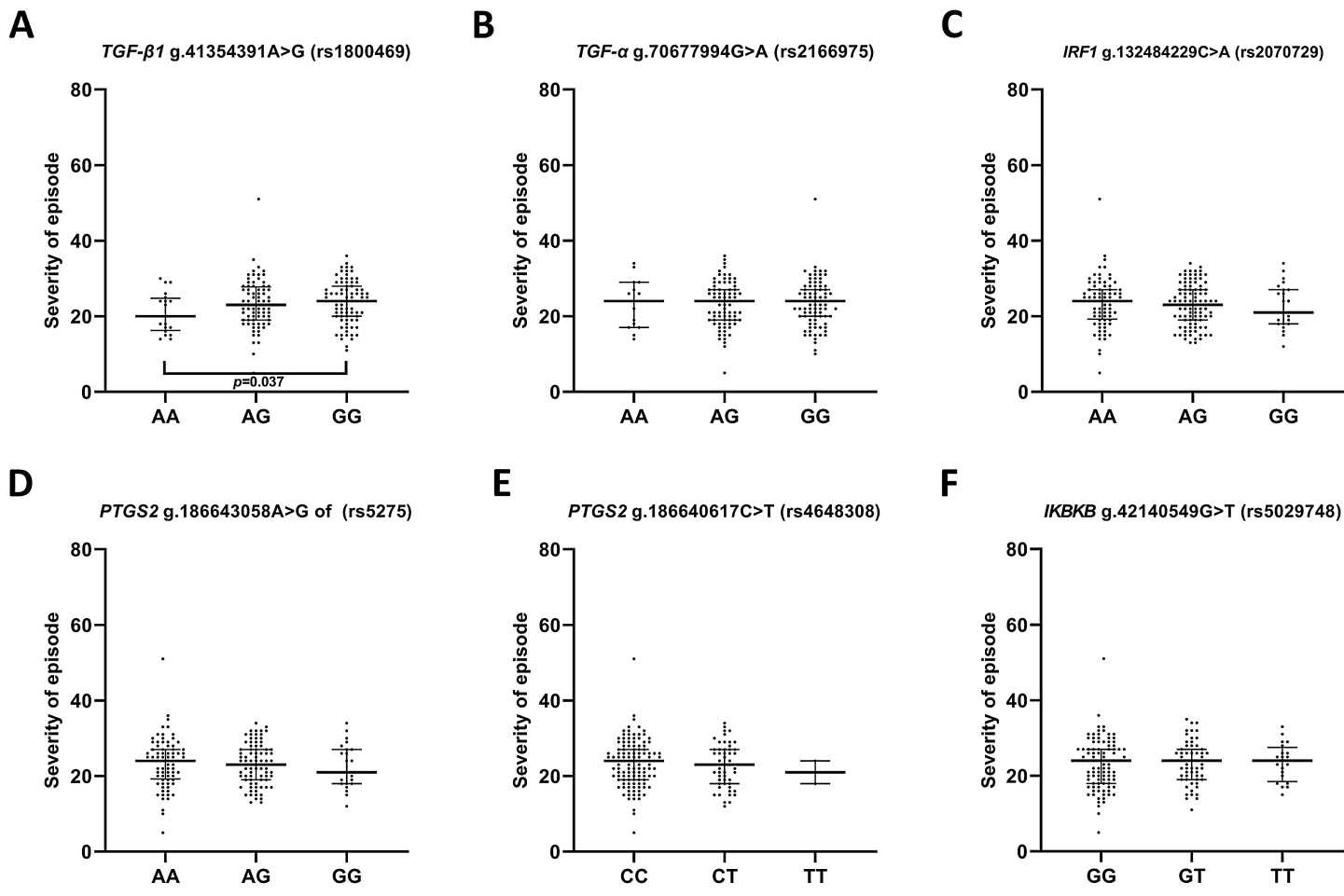

**Figure 2 Distribution of the severity of episode (before therapy) and single nucleotide polymorphisms localized in inflammatory genes.** Severity of current episode according to 21-item Hamilton Depression Rating Scale (HAM-D) (A) *TGFB1* g.41354391A>G (rs1800469) (B) *TGFA* g.70677994G>A (rs2166975) (C) *IRF1* g.132484229C>A (rs2070729) (D) *PTGS2* g.186643058A>G (rs5275) (E) *PTGS2* g.186640617C>T (rs4648308) (F) *IKBKB* g.42140549G>T (rs5029748). Results are presented as scatter dot plots. The horizontal lines denote the median, while the whiskers show the inter-quartile range.

Val159Val. This terminal amino acid is present in the precursor protein and is necessary for glycosylation during protein maturation as well as protein localization to the cell surface (*Briley et al., 1997*). In our study, we were the first to show a link between rs2166975 polymorphism of *TGFA* and depression. The results confirmed that A/G genotype of rs2166975 is more frequently distributed in patients suffering from depression. Interestingly, the same genotype increased the risk of MDD only in man population. In the case of the gene–gene interactions between polymorphism of *TGFA* and other SNPs, analysis confirmed that A/G-A/C combined genotypes of rs2166975—*TGFA* and rs207072—*IRF1* are associated with higher chance to develop MDD. In addition, A/G-G/G genotypes of rs2166975—*TGFA* and rs5275—*PTGS2* is associated with higher risk of MDD, while G/G-G/G homozygotes decreased this chance. It was indicated that rs2166975 in *TGFA* gene, showed association with the risk of cleft palate (*Morkūnienė et al., 2007*). Furthermore, another study confirmed, using transmission disequilibrium test, that minor allele of

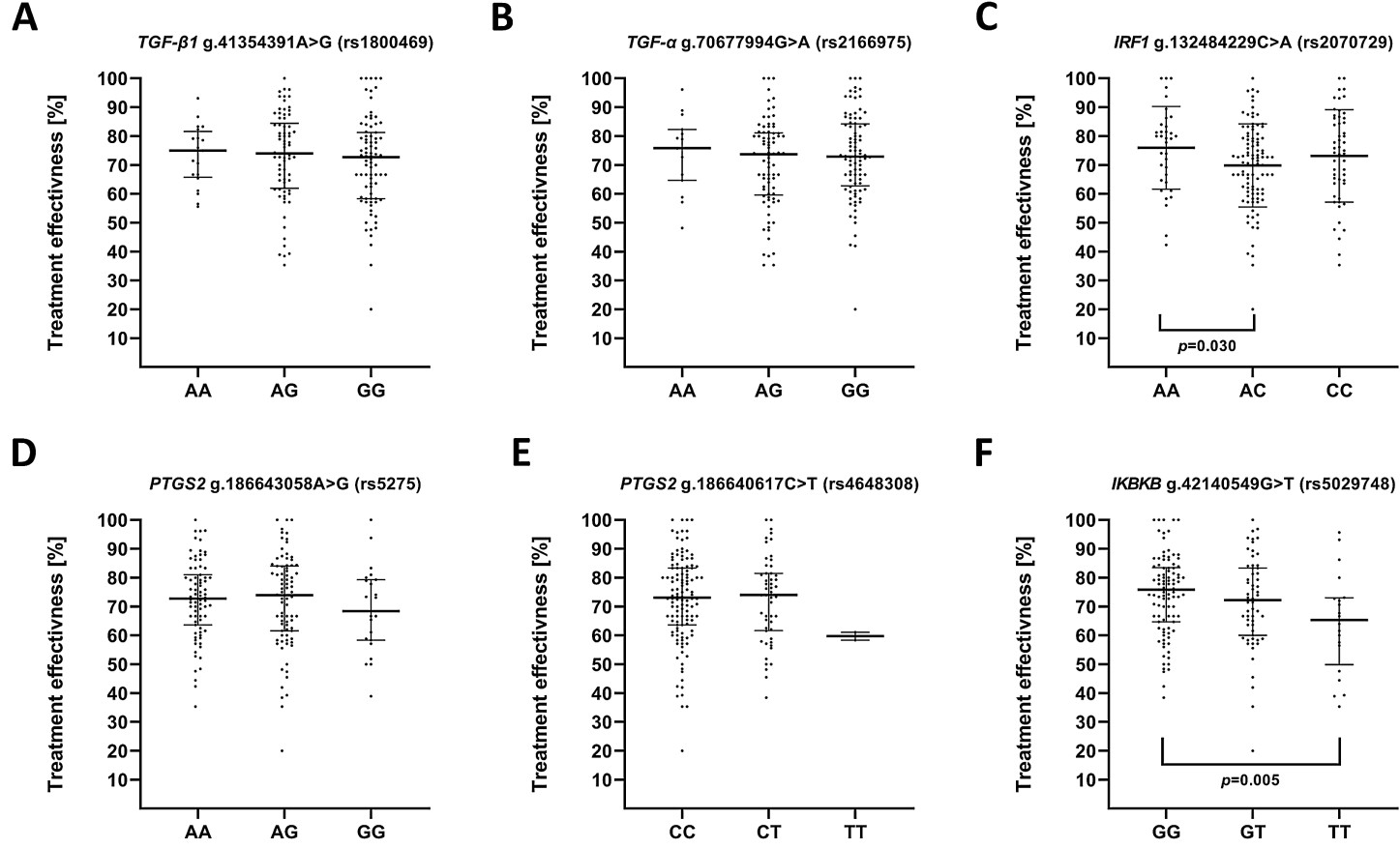

**Figure 3** **Impact of single-nucleotide polymorphisms localized in inflammatory genes on the effectiveness of the treatment.** Treatment effectiveness expressed as percentage of HAM-D decline after therapy. (A) *TGFB1* g.41354391A>G (rs1800469) (B) *TGFA* g.70677994G>A (rs2166975) (C) *IRF1* g.132484229C>A (rs2070729) (D) *PTGS2* g.186643058A>G (rs5275) (E) *PTGS2* g.186640617C>T (rs4648308) (F) *IKBKB* g.42140549G>T (rs5029748). Results are presented as scatter dot plots. The horizontal lines denote the median, while the whiskers show the inter-quartile range.

rs2166975 was over-transmitted to cleft-palate cases (*Carter et al., 2010*). Although there are no studies investigating role of rs2166975 polymorphism in depression or any other psychiatric disorders, our results suggest important role of investigated polymorphism in pathophysiology and course of depression.

The second studied SNP, g.41354391A>G (rs1800469) of *TGFB1*, is located on 19q13.2 in the proximal negative regulatory region of the gene. The human TGFB1 protein is considered to be one of the immunosuppressive cytokines, which plays crucial role in CNS development (*Sousa Vde et al., 2004*). It is responsible for such functions as astrocyte differentiation, synaptogenesis and neuronal migration (*De Sampaio e Spohr et al., 2002*; *Sousa Vde et al., 2004*; *Feng & Ko, 2008*; *Siegenthaler & Miller, 2004*). Our results show that rs1800469 polymorphism is associated with both severity of depressive episodes and age of the onset of the disease. Precisely, carriers of G/G genotype are characterized by more severe episodes than A/A genotype carriers, which may correlate with increased concentrations of TGFB1. Moreover, a significant difference in the age of the first episode of MDD was found between A/A and A/G genotypes, as well as A/A and G/G genotypes

of rs1800469—*TGFB1*. In accordance to our findings, TGFB levels were found to be increased in people suffering from MDD (*Davami et al., 2016*; *Kim et al., 2007*; *Kim et al., 2008*) as well as in Chronic HBV-Infected Patients (CHB) with mild depression symptoms (*Bahramabadi et al., 2017*). It has been reported that rs1800469, is not associated with neither Alzheimer's disease risk (*Chang et al., 2013*) nor Schizophrenia (*Kapelski et al., 2015*). However rs1800469 of *TGFB1* is associated with altered plasma levels of TGFB1, which may modulate a susceptibility to MDD (*Shah et al., 2006*; *Wang et al., 2008*). Data suggest that, allele G is associated with lower expression of *TGFB1* (*Shah et al., 2006*). On the other hand, another study confirmed that genotypes A/G and G/G was correlated with increased plasma TGFB1 concentrations, indicating that G allele is associated with higher production of the protein (*Wang et al., 2008*). It was found that other SNPs of *TGFB1* could be associated with MDD. In the case of rs1800470 (codon 10), genotype T/T is significantly more frequently distributed in depressed patients (*Mihailova et al., 2016*). Moreover, another study revealed that C/C genotype of the same SNP is positively correlated with higher risk of depression development and more severe episodes of the disease (*Caraci et al., 2012*). Although TGFB1 is considered to play important role in psychoneuroimmunology, there is only few research about its association with mental disorders, and interestingly there is no other studies investigated role of mentioned rs1800469 in MDD.

In our study we also investigated whether SNPs in *PTGS2* gene are involved in MDD development. As mentioned in Introduction, PTGS2 participates in inflammatory processes partly related with neurodegeneration in CNS (*Minghetti, 2004*). There is evidence demonstrating that rs20417 polymorphism of *PTGS2* may play a role in MDD. Precisely, presence of G allele is strongly associated with increased risk of depression development (*Gałecki et al., 2010*). However, we have not included this polymorphism in our study. Instead, we explored g.186640617C>T (rs4648308) polymorphism located on 1q31.1. There are evidence of its involvement in depression. Precisely, allele T and C/T genotype (in positive strand allele A and G/A genotype) of mentioned SNP are associated with significantly increased risk of IFN-α-induced depression (*Su et al., 2010*). Part of our result are consistent with this findings, namely, we found that C/T heterozygote increased risk of MDD in woman, as well as the C allele increased this chance in man group. On the contrary, we also reported that T/T genotype carriers of this SNP are less likely to develop depression in general population. Similarly, in man group allele T was also negatively correlated with depression prevalence.

Second polymorphism of *PTGS2* gene, g.186643058A>G (rs5275) located on 1q31.1, is a functional SNP, which modulates expression of PTGS2. We were first to found that allele G is connected with higher chance of MDD occurrence. Additionally, it is confirmed that this SNP is associated with severe pain in lung cancer patients. Namely, A/A and A/G (in forward strand T/T and T/C) carriers experience more severe pain than G/G carriers (*Reyes-Gibby et al., 2009*; *Reyes-Gibby et al., 2013*). However, *Mendlewicz et al. (2012)* found no association between *PTGS2* rs5275 polymorphism and treatment response and remission of MDD. Still, there are no other studies investigated aforementioned SNPs in *PTGS2* gene in context of MDD.

Another SNP candidate in our research was g.42140549G>T (rs5029748) of *IKBKB* gene. It is located on 8p11.21, in intronic region of the gene, thus do not cause amino acid substitution. We were first to analyze the mentioned polymorphism as a risk factor for MDD. Our main finding relates to the connection between this SNP and effectiveness of depression treatment. Namely, we demonstrated differences in SSRI response between carriers of G/G and T/T genotypes. Moreover, presence of G/T genotype of rs5029748 is associated with increased risk of MDD development either in general or man population, while the T/T homozygote of the same gene variant reduces this risk in the same studied groups. In addition, carrier of combined G/T-A/G genotypes of rs5029748— *IKBKB* and rs5275—*PTGS2* are more likely to develop MDD, while T/T-G/G genotype showed protective effect. Moreover A/G-G/T genotype of rs5029748 *IKBKB* and rs2166975—*TGFA*, increased risk of depression but G/T-T/T are associated with lower risk of disease. The trend of increasing risk of depression prevalence is also present in the case of linked genotypes of rs5029748—IKBKB and rs4648308—PTGS2. Some studies revealed association between aforementioned SNP and risk of colorectal or colon cancer (*Seufert et al., 2013*; *Curtin et al., 2013*). Precisely, minor allele T of rs5029748, was associated with decreased risk of colon cancer (*Curtin et al., 2013*). Although our result showed that single-nucleotide polymorphism of *IKBKB* may play significant role in MDD, they have not been investigated in pathogenesis of the disease before.

The g.132484229C>A (rs2070729)—*IRF1* polymorphism was the last studied SNP in this article. It is located on 5q31.1 in intronic gene region. The SNP is associated with susceptibility to hepatitis C virus (HCV) infection (*Fortunato et al., 2008*). What is more, allele C of this SNP is linked to higher vulnerability HIV-1 acquisition (*Lingappa et al., 2011*). To our best knowledge, we were first to analyze role of rs2070729 in MDD. Regarding the effect of investigated SNP on treatment efficiency, data in our study showed significant differences in antidepressant response between A/A and A/C genotypes of rs2070729—*IRF1*, A/A carriers were more likely to better treatment response. Exact explanation of this mechanism has not been elucidated yet in previous research. However, since A allele is a minor one in European population, we speculate that it might be associated with decreased expression of IRF1 and thus reduction of inflammatory cytokine release. Therefore, together with anti-inflammatory properties of antidepressants it could enhance the their effect. We also found that carriers of A/C genotype of rs2070729— *RF1* were linked with A/G of rs5275—*PTGS* or G/T of rs5029748—*IKBKB* had a greater risk of MDD appearance. These results suggest that SNP in *IRF1* gene may have impact in depression development.

Our preliminary study has several potential limitations. Firstly, the sample size was relatively small. Nevertheless, two resampling approaches were performed so as to minimize the risk of obtaining false positive results. Another limitation was the homogenic ethnicity of studied group. This could reduce the potential to extrapolate the results to other ethnic groups. Furthermore, it must be emphasized that there is limited data on the impact of these SNPs on the level of mRNA and protein expression/activity. Consequently, presented results should be considered preliminary and interpreted with caution.

## CONCLUSIONS

The single-nucleotide polymorphisms located in *IRF1, IKBKB, TGFA, TGFB1, PTGS2* genes modulate the risk of occurrence, age of onset, severity of the disease and response to the antidepressant treatment. Our result suggest that inflammatory pathways, in which studied genes are involved may be at least partially implicated in etiology of MDD. Moreover, discovery about impact of *IRF1* and *IKBKB* SNPs on treatment response could contribute to the discovery of effective, personalized pharmacotherapy. However, future studies should elucidate the implication of the studied polymorphisms in biological functions, for example, mRNA and protein expression, protein activity. On the whole, our results might cast a new light on the pathogenesis of major depressive disorders.

### Funding

This study was supported by funding from a scientific research grant from the Polish National Science Centre (No. UMO-2015/19/BNZ7/00410). The funders had no role in study design, data collection and analysis, decision to publish, or preparation of the manuscript.

### Grant Disclosures

The following grant information was disclosed by the authors:
Polish National Science Centre: UMO-2015/19/BNZ7/00410.

### Competing Interests

The authors declare that they have no competing interests.

### Author Contributions

- Katarzyna Bialek conceived and designed the experiments, performed the experiments, analyzed the data, prepared figures and/or tables, authored or reviewed drafts of the paper, and approved the final draft.
- Piotr Czarny conceived and designed the experiments, performed the experiments, analyzed the data, prepared figures and/or tables, authored or reviewed drafts of the paper, and approved the final draft.
- Cezary Watala analyzed the data, prepared figures and/or tables, authored or reviewed drafts of the paper, and approved the final draft.
- Paulina Wigner conceived and designed the experiments, performed the experiments, authored or reviewed drafts of the paper, and approved the final draft.
- Monika Talarowska performed the experiments, analyzed the data, authored or reviewed drafts of the paper, diagnosis of the patients, and approved the final draft.
- Piotr Galecki conceived and designed the experiments, performed the experiments, authored or reviewed drafts of the paper, diagnosis of the patients, and approved the final draft.
- Janusz Szemraj conceived and designed the experiments, analyzed the data, authored or reviewed drafts of the paper, and approved the final draft.
- Tomasz Sliwinski conceived and designed the experiments, analyzed the data, authored or reviewed drafts of the paper, and approved the final draft.

## Human Ethics

The following information was supplied relating to ethical approvals (i.e., approving body and any reference numbers):

Protocol of the study was approved by the Bioethics Committee of the Medical University of Lodz (No. RNN/70/14/KE).

## Data Availability

The raw data is available in the Supplemental Files.

## Supplemental Information

Supplemental information for this article can be found online at http://dx.doi.org/10.7717/peerj.8676#supplemental-information.

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
