# Peer review of "Novel association between TGFA, TGFB1, IRF1, PTGS2 and IKBKB single-nucleotide polymorphisms and occurrence, severity and treatment response of major depressive disorder"

_PeerJ, doi:10.7717/peerj.8676_

## Round 0.1 · original submission · Major Revisions

Please address as many comments as possible as this will improve the impact of your study.

Reviewer 1 ·

Basic reporting

Please see the comments in the next section.

Experimental design

This study was designed to determine six SNPs in inflammatory-related genes associated with MDD occurrence, age of onset, and response to SSRI treatment.

Major comments and questions:
This study included 180 MDD subjects and 180 healthy controls. The subjects were selected randomly. However, the authors need to clarify whether this is a sex-age-matched control study. This is critical, because the study was not only designed to determine the “association” with the disease risk, but also study the age, and sex effects on the occurrence of the disease and treatment response. It is a population study, therefore, this is crucial to clarify the study subjects’ characteristics. In addition, this will be a very important factor for data interpretation. In fact, 2/3 of patients with MDD are women, virtually in every MDD study.

The rationale of selecting these SNPs should be demonstrated in the manuscript. It should not just because none of these SNP haven been studied in depression (line 275). In fact this is not a fair statement. A series of large MDD GWAS have been published over the years, and each of these immune gene might have implication in MDD pathophysiology or SSRI treatment response. One of the major critiques for this manuscript is the selection of SNPs. In addition, it remains to be determined whether the findings can be replicated. Furthermore, this study does not provide any functional genomic data to show the biological impacts of the SNPs which reveal positive results in their study population. Therefore, this is a purely “association study” in a small population which fails to provide meaningful biology beyond the p values. I suggest that the authors should take advantage of published data ie, Nature Neuroscience volume 22, pages343–352(2019), or STAR*D study…etc which can be served as a replication study.
Finally, the conclusion seems overstatement and fails to be supported by their finding (line377). “discovery about impact of selected SNP on treatment response could contribute to the selection of effective, personalized pharmacotherapy. On the whole, our results cast a new light on the pathogenesis of major depressive disorders”.

Line 181: Please clarify the statistic methods for data analysis. Why is that the authors used unconditional multiple logistic regression model to determine the association of the SNPs and the occurrence of MDD? What are they co-variants in these analyeis?

Line 264: treatment efficacy. HAMD scores at baseline and 8 week treatment should be presented in the manuscript. In addition, with regard to treatment efficacy, it will be helpful to define as “ response (reduction from baseline of ≥ 50% in the total score) and remission (total HAMD-17 score ≤ 7), all of which seem more standard definition for treatment outcome.

Validity of the findings

Line 223: gene –gene interaction. These SNPs are within immune genes, and in differently inflammatory signaling pathways. I am not sure how the authors used one SNP in a gene to performed gene gene interaction and if these results are valid?

Additional comments

Minor comments and questions:

Line 31: standard nomenclature should be used throughout the manuscript. ie TGFA, TGFB
Line 37: a gene-gene analysis revealed a link between studied polymorphisms and depression. Please clarify if the analytical method is valid.
Lin83: od
Table 1: please correct and specify the location of each SNP included in the study, ie, which exon? If it’s located in 5’ of the gene, it should be 5’ of X gene. ..etc. please use standard way to express gene structure. It will be helpful for the readers if this table can include minor allele frequency for each SNP in European population from public database ie NCBI SNP!

Reviewer 2 ·

Basic reporting

In the present manuscript, authors evaluated the association of six polymorphisms in TGF-β1, IRF, PTGS2, TGF-α and IKBKB with depression, and antidepressant efficacy, in a total of 360 (180 patients and 180 controls) DNA samples, using TaqMan probes. Authors concluded that the studied SNPs might influence the risk of occurrence, age of onset, severity of depression and response to the antidepressant treatment.

#1 In general, the paper is easy to read, however it would benefit from a global typos, grammar and punctuation revision. E.g. lines 53, 93,96, 103, 123, 273.

#2 Background information is relevant, pertinent and allows understanding of the context. However, several sentences lack references. Please see lines 56 and 57 for instance. Additionally, reference 1 is incomplete.

#3 Structure is, in general, in agreement with Peer J guidelines.

#4 Regarding raw data, authors kindly provided it in the supplementary files 1.

Experimental design

Research is within scope of the journal, and research questions are well defined and relevant. However, some methods need a better clarification/to be completed:

#1 Were depressed patients treated with the same SSRI (line 146)? It is not clear in the text.

#2 In Methods section authors say that DNA purity was controlled by acquiring absorbance ratio at 260 nm and 280 nm (lines 165-166). In which equipment was it determined?

#3 Since sample is not large, as referred by the authors, paper would benefit if the power of the study could be calculated.

#4 Authors calculate treatment efficacy thought a formula, which is display in results section (lines 262-263). It is not clear why they use this formula to calculate efficacy. Was it based in other study? The formula must be displayed in methods section, not in results section.

#5 An extensive statistical analysis was performed, however, and since six polymorphisms in TGF-β1, IRF, PTGS2, TGF-α and IKBKB were evaluated, the paper would benefit if correction for multiple testing (such as Bonferroni) could be performed.

#5 Figures and tables are relevant and provide with the quality, however tables 3- 4, due to its extension, would benefit if sent to supplementary materials.

Validity of the findings

Discussion is extensive, however I fell it needs a bit more detail. I suggest that you improve the justification for the impact of the studied polymorphisms with depression/antidepressant response, to provide a clearer justification for your study.

#1 For example for TGF α and TGF β, I suggest you include the following papers to improve you discussion:
https://www.ncbi.nlm.nih.gov/pubmed/27049572
https://www.ncbi.nlm.nih.gov/pubmed/29237050

#2 In line 362, regarding IRF1 gene, authors concluded that AA carriers had a better treatment response. It is not clear why. Authors could provide a putative mechanism, supported in other studies, for these results.

---

## Round 0.2 · accepted · Accept

The reviewer's comments have been all addressed.